# Tuning Multi-mode Token-level Prompt Alignment across Modalities

**Dongsheng Wang, Miaoge Li, Xinyang Liu, MingSheng Xu, Bo Chen***
School of Electronic Engineering, Xidian University, Xi'an, China
`{wds,limiaoge,xinyangatk,msxu}@stu.xidian.edu.cn`
`bchen@mail.xidian.edu.cn`

**Hanwang Zhang**
School of Computer Science and Engineering, Nanyang Technological University, Singapore
`hanwangzhang@ntu.edu.sg`

## Abstract

Advancements in prompt tuning of vision-language models have underscored their potential in enhancing open-world visual concept comprehension. However, prior works only primarily focus on single-mode (only one prompt for each modality) and holistic level (image or sentence) semantic alignment, which fails to capture the sample diversity, leading to sub-optimal prompt discovery. To address the limitation, we propose a multi-mode token-level tuning framework that leverages the optimal transportation to learn and align a set of prompt tokens across modalities. Specifically, we rely on two essential factors: 1) multi-mode prompts discovery, which guarantees diverse semantic representations, and 2) token-level alignment, which helps explore fine-grained similarity. Consequently, the similarity can be calculated as a hierarchical transportation problem between the modality-specific sets. Extensive experiments on popular image recognition benchmarks show the superior generalization and few-shot abilities of our approach. The qualitative analysis demonstrates that the learned prompt tokens have the ability to capture diverse visual concepts. The code is available at https://github.com/wds2014/ALIGN.

## 1 Introduction

Recently, prompt tuning has experienced significant advancements in adapting large pre-trained vision language models (PVLs) such as CLIP [1] and BLIP [2] to downstream tasks [3–6]. A typical PVL model consists of two branch networks: the text and image encoders. These networks are used to extract the corresponding modality features. PVLs are often contrastively pre-trained on Web-scale image-text pairs, which encourage the alignment of visual concepts with natural language in the shared semantic space. One of the core ideas behind prompt tuning is to formulate the downstream tasks into the original pre-training pipeline. For example, CLIP designs category descriptions with a manual prompt template "*a photo of a* $\{class\}$", which works well in generic image recognition. Unlike fine-tuning, where the entire model is tuned using task-specific objectives, demands prohibitive computing cost, and poses a risk of knowledge shift issues [7–9], prompt tuning fixes the model parameters instead and optimizes prompt vectors, which act as demonstrations to help extract task-related features. This significantly benefits the representations via PVLs, even in performing zero-shot inference without training samples.

However, identifying optimal prompts for PVLs is not a trivial task, which usually needs to solve the intricate semantic alignments between the textual and visual modalities. Inspired by the success

---

*Corresponding author

37th Conference on Neural Information Processing Systems (NeurIPS 2023).

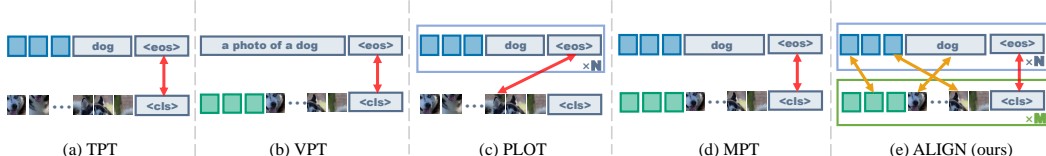

Figure 1: The alignment comparison in recent prompt tuning methods. The proposed ALIGN learns multi-modal multi-mode prompts at the same time, resulting in comprehensive alignments.

of prompt learning in neural language models (NLP) [10, 7, 11], approaches called textual prompt tuning (TPT) are proposed to learn continuous prompt embeddings for CLIP's text encoder, *e.g.*, "*X X X X {class}*", where "X" denotes the learnable vectors [3, 4]. Optimized with a task-specific loss, the learned prompt embeddings distill the pre-trained knowledge encoded in the fixed parameters, achieving better flexibility and efficiency than hand-crafted methods [1]. To improve the generalization of TPT on unseen classes, many studies attempt to give the solutions from gradient flow [12, 13], prototype and composition prompt learning [14–16]. Moving beyond learning a single-mode prompt, which often fails to capture diverse concepts, various methods prefer to explore multiple prompts based on ensemble learning[1], optimal transport [17] and Bayesian inference [18–20], showing robust alignments and better performance.

In parallel with TPT, visual prompt tuning (VPT) focuses on the patch embedding space of the CLIP's image encoder [6]. VPT views images as a patch sequence and introduces visual prompts to enhance the image representations, *e.g.*, "*X X X X {image}*", where "image" denotes the image patch sequence. VPT provides a simple and efficient idea to extract task-relevant visual features, which has been widely applied to many visual tasks, for example, video understanding [21], domain adaptation [22], transfer learning [23] and image segmentation [24–26]. More recently, there has been a research trend to combine TPT and VPT to learn multi-modal prompts together [27, 28]. However, they currently concentrate on single-mode prompt discovery, *i.e.*, only one prompt for one modality, which may be insufficient to represent a class [17]. This issue is even more acute in multi-modal prompt learning, where both visual and textual concepts and their alignments need to be inferred. Additionally, it is less sound to represent the image and label only with the global features [29, 30], which could lose local region features of the target object, resulting in sub-optimal classification.

To this end, this work develops a comprehensive prompt tuning framework, where multi-modal multi-mode prompts are learned by building the prompt and token-level optimal transport (OT). Formally, after feeding multiple prompt inputs into the modality-specific encoder, our prompt-level OT views each image as a discrete distribution $P$ over the visual prompt space and views each label as a discrete distribution $Q$ over the textual prompt space. With such formulation, the classification task becomes to measure the distance between $P$ and $Q$. Moreover, along with the global prompt-level features, the patch (or token) embeddings capture the local region features of the target object (or category description). This motivates the token-level OT, where each prompt output is modeled as a discrete distribution over the token embedding space. The cost matrix is then calculated between the visual patches and textual tokens, enabling the token-level alignments. Crucially, the cost matrix in prompt-level OT that measures the transport cost between prompts from two domains is now converted to integrate the global features and the output of the token-level OT. This hierarchical connection makes it possible to predict the label with the detailed token and patch features, resulting in higher accuracy.

In summary, our method provides a novel prompt tuning framework that incorporates multiple modalities and token-level alignments via the hierarchical OT. The prompt-level OT learns the diverse semantics of a class from both image and language domains, and the token-level OT explores fine-grained alignments between token embeddings. Notably, with different hyperparameter settings, the variants of the proposed model cover many previous works, offering flexibility for easy adaptation across diverse applications. The main contributions of the paper are as follows:

- We propose a multi-mode token-level alignment framework for multi-modal prompts tuning, where multiple prompts are learned to improve the representation for both visual and textual modalities. With special settings, many previous works can be margined into our framework.

- We formulate the prompt tuning task as the distribution matching problem, and develop the prompt and token-level OT to tackle the task with a principle and elegant solution.
- We apply our method to few-shot classification, dataset transfer learning and domain generalization. Experiential results on widely used datasets show the superiority of the proposed model.

## 2 Background

### 2.1 Multi-modal Prompt Tuning

Multi-modal prompt tuning (MPT) [28, 27] is a newly developed task that enables the joint learning of textual and visual prompts for PVLs. Instead of optimizing the unimodal prompts separately, the joint tuning paradigm not only leverages the two branch networks of PVLs, but also allows interactions between two modalities during training, resulting in dynamic alignments. Without loss of generality, we use a vision transformer (ViT) based CLIP for example, which consists of a ViT as an image encoder $f$ and a transformer as a language encoder $g$. Given an input image $\mathbf{X} \in R^{H \times W \times 3}$ and $K$ label names $\{class_k\}_{k=1}^K$. MPT first incorporates $b$ learnable tokens as visual prompts $\{\boldsymbol{v}_i \in R^{d_v}\}_{i=1}^b$, and another set of $b$ learnable tokens as textual prompts $\{\boldsymbol{t}_i \in R^{d_l}\}_{i=1}^b$. After concatenating them alongside the image patches and class names, one can obtain the output of CLIP as:

$$[\boldsymbol{z}, \widetilde{\boldsymbol{e}}_1, ..., \widetilde{\boldsymbol{e}}_O, \widetilde{\boldsymbol{v}}_1, ..., \widetilde{\boldsymbol{v}}_b] = f(<cls>, \boldsymbol{e}_1, ..., \boldsymbol{e}_O, \boldsymbol{v}_1, ..., \boldsymbol{v}_b),$$
$$[-, \widetilde{\boldsymbol{t}}_1, ..., \widetilde{\boldsymbol{t}}_b, \widetilde{\boldsymbol{w}}_{k,1}, ..., \widetilde{\boldsymbol{w}}_{k,k_l}, \boldsymbol{h}_k] = g(<cls>, \boldsymbol{t}_1, ..., \boldsymbol{t}_b, \boldsymbol{w}_{k,1}, ..., \boldsymbol{w}_{k,k_l}, <eos>),$$

where $<cls>, <eos>$ are virtual tokens, $[\boldsymbol{e}_1, ..., \boldsymbol{e}_O]$ are $O$ image patch embeddings, and $[\boldsymbol{w}_{k,1}, ..., \boldsymbol{w}_{k,k_l}]$ are token embeddings with length $k_l$ of $k$-th class. After the stacked self-attention layers of $f$ and $g$, CLIP outputs the token embeddings and views $\boldsymbol{z}$ and $\boldsymbol{h}_k$ as the prompt-level features of the image and label, respectively. Empirical findings suggest that it is more effective to obtain the vision prompt $\boldsymbol{v}$ by projecting the language prompt $\boldsymbol{t}$ through a vision-to-language mapping function, such as $\boldsymbol{v} = F(\boldsymbol{t})$, rather than learning them independently [28, 6]. Finally, MPT estimates the label of $\boldsymbol{x}$ according to the cosine similarity score:

$$p(y = k|\boldsymbol{x}) = \frac{\exp(\text{sim}(\boldsymbol{z}, \boldsymbol{h}_k)/\tau)}{\sum_{k'=1}^K \exp(\text{sim}(\boldsymbol{z}, \boldsymbol{h}_{k'})/\tau)}, \tag{1}$$

where $\tau$ is the fixed temperature parameter. MPT unifies the ideas of TPT and VPT by directly tuning the visual prompts $\boldsymbol{v}$ and textual prompt $\boldsymbol{t}$ at the same time. Eq. 1 indicates that the text encoder $g$ takes the category prompts as input and outputs $\boldsymbol{h}$, which serves as the corresponding classifier weights. Thanks to the pre-trained knowledge in CLIP, MPT retains the ability to perform open-set classification. Note that both the encoders $f$ and $g$ in CLIP are frozen and only the prompt sequences $\boldsymbol{v}$ and $\boldsymbol{t}$ are optimized during downstream training. This process can be seen as a bootstrapping step that helps guide the encoders to extract task-relevant features.

### 2.2 Optimal Transport Distance

Optimal transport (OT) is an efficient tool to measure the distance between two distributions, which is widely used in recent machine learning studies, such as text analysis [31–33], computer vision [34–39] and generative model [40, 41]. Here we review the discrete OT matching and refer readers to [42] for details. Given two sets of data points $\mathbf{X} = \{x_i\}_{i=1}^m$ and $\mathbf{Y} = \{y_j\}_{j=1}^n$, of which discrete distributions are formulated as $p = \sum_{i=1}^m a_i \delta_{x_i}$ and $q = \sum_{j=1}^n b_j \delta_{y_j}$, respectively. $\boldsymbol{a} \in \Delta^m$ and $\boldsymbol{b} \in \Delta^n$, where $\Delta^m$ denotes the probability simple of $R^m$. We define the cost matrix between $\mathbf{X}$ and $\mathbf{Y}$ as $\mathbf{C} = (C_{ij}) \in R_{\geq 0}^{m \times n}$, where $C_{ij} = c(x_i, y_j)$ is the transport cost from $x_i$ to $y_j$, with $c$ is the cost function. The goal of OT is to optimally transport $p$ to $q$ at the smallest cost:

$$d_{\text{OT}}(p, q; \mathbf{C}) := \min_{\mathbf{T} \in \Pi(p,q)} <\mathbf{T}, \mathbf{C}>, \tag{2}$$

where $< \cdot, \cdot >$ denotes the Frobenius dot-product and $\mathbf{T} \in \mathbb{R}_{>0}^{m \times n}$ denotes the transport plan to be learned. OT distance is then minimized over all the joint probabilities of $m \times n$ space with two marginal constraints $\Pi(p, q) := \{\mathbf{T} : \mathbf{T}\mathbb{1}_n = \boldsymbol{a}, \mathbf{T}^T\mathbb{1}_m = \boldsymbol{b}\}$, where $\mathbb{1}_m$ denotes m-dimensional

all-one vector. As directly learning the optimal plan $\mathbf{T}$ in Eq. 2 can be time-consuming for large-scale problems, Sinkhorn distance from [42, 43] introduces the entropic constraint on the transport plan $h(\mathbf{T}) = \sum_{m,n} -T_{mn}\ln(T_{mn})$ and thus the resulting algorithm estimates $\mathbf{T}$ within a few iterations, showing better flexibility and scalability.

# 3 The Proposed Model

## 3.1 Overall Method

In this section, we introduce the technical details of our proposed model, named `ALIGN`, a holistic framework for multi-modal prompt tuning with optimal transport (shown in Fig. 2). Benefiting from the carefully designed multi-mode token-level alignment module, most existing works can be merged into our `ALIGN` with special settings. Intuitively, humans learn one class with various concepts, which provides sufficient semantic features, such as color, layout, and shape, to distinguish it from others [17]. Inspired by this, one of the goals of this work is to learn $M$ visual prompt and $N$ textual prompt simultaneously. Specifically, we first introduce our prompt-level OT, where each image and label are modeled as the discrete distributions $P$ and $Q$ over $M$-dimensional visual space and $N$-dimensional textual space. Moreover, instead of representing the prompt outputs as a single point, *e.g.*, the global features $\boldsymbol{z}$ and $\boldsymbol{h}$, we distill the token-level knowledge implied in CLIP. Recalling that, the $n$-th textual prompt output of the $k$-th class contains $b + k_l$ token embeddings and the $m$-th visual prompt output of an image contains $b + O$ patch embeddings, which capture the local-region features of corresponding modalities. This motivated us to develop the token-level OT that makes token-level comparisons for fine-grained alignments. As a result, $m$-th and $n$-th points in $P$ and $Q$ themselves are further modeled as discrete distributions over the shared token embedding space. Due to the compelling two-level OT connections, where the cost matrix in prompt-level OT is obtained by the output of token-level OT, the learned transport plan captures both the prompt and token-level features, which provides a principled and elegant way to estimate the distance between label and image sets.

## 3.2 Multi-mode Token-level Prompt Alignment

Moving beyond MPT which learns a single-mode prompt to describe the class and estimates the similarity based on prompt-level features, we aim to explore multi-mode representations in the textual and visual domains and make fine-grained alignment to improve the prediction accuracy. Now we have $M$ groups of visual prompts $\{\boldsymbol{v}^m\}_{m=1}^M$ and $N$ groups of textual prompts $\{\boldsymbol{t}^n\}_{n=1}^N$, where each $\boldsymbol{v}^m \in R^{d_v \times b}$ and $\boldsymbol{t}^n \in R^{d_l \times b}$ are learnable prompt sequences with length $b$. Mathematically, we employ two empirical distributions $P$ and $Q$ to model the sets of two modalities:

$$P = \sum_{m=1}^M \frac{1}{M}\delta_{\boldsymbol{x}_m}, \qquad Q = \sum_{n=1}^N \frac{1}{N}\delta_{\boldsymbol{y}_n}, \tag{3}$$

where $\boldsymbol{x}_m$ and $\boldsymbol{y}_n$ denote the $m$-th visual output and $n$-th textual output in the $d$-dimensional latent space. They are further modeled as discrete distributions over token-level embeddings, which will be introduced later. Eq. 3 views each prompt equally and adopts the uniform distribution to model the weights. With those two semantic sets $P$ and $Q$, the distance between images and labels is no longer calculated by first representing each image and label as a single point and then using the cosine similarity. `ALIGN` prefers to mine multi-mode features to describe various class concepts, resulting in better representations. The distance thus can be formulated as an entropy-regularized prompt-level OT problem [42]:

$$d_{\text{OT}}^\lambda(P, Q; \mathbf{C}) := d_{\text{OT}}(P, Q; \mathbf{C}) - \lambda h(\mathbf{T}) \tag{4}$$

where $\lambda > 0$ is the weight of regularization, and $\mathbf{C} \in R^{M \times N}$ is the cost matrix between visual set $\boldsymbol{x}$ and textual set $\boldsymbol{y}$. $\mathbf{T} \in R^{M \times N}$ is the to-be-learned transport plan with the marginal constraint, *e.g.*,$\mathbf{T}\mathbb{1}_N = 1/M, \mathbf{T}^T\mathbb{1}_M = 1/N$. Note that, $T_{mn}$ measures the transported probability from $m$-th visual prompt to $n$-th textual prompt, and a large value means the high semantic connection between two prompts across modalities. Therefore, Eq. 4 estimates the expected transport cost between $P$ and $Q$, which provides a principle solution to calculate the similarity between the images and labels.

Noticeably, the cost matrix $\mathbf{C}$ in Eq. 4 plays a critical role in the learning of $\mathbf{T}$, and intuitively, the larger the transport costs between two points are, the lower the transport probabilities will be. A

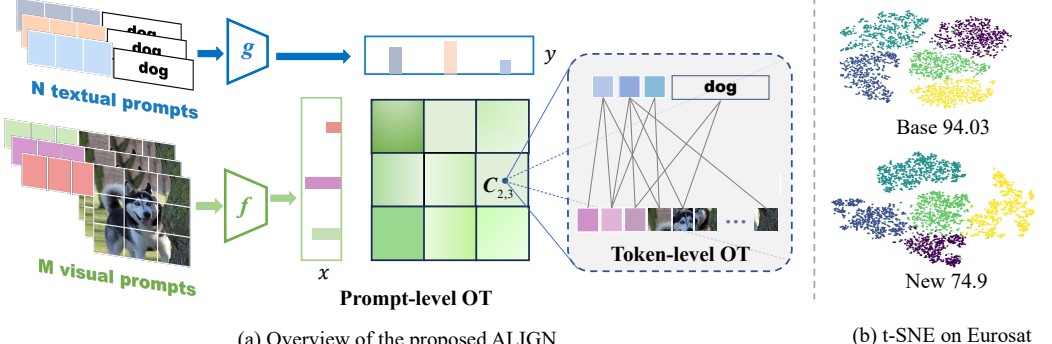

(a) Overview of the proposed ALIGN

(b) t-SNE on Eurosat

Figure 2: (a) The framework of the proposed `ALIGN`. `ALIGN` learns multiple prompts for PVLs by aligning modality-specific distributions with hierarchical OT. (b) The t-SNE visualization of image embeddings of `ALIGN`.

natural choice is to specify $\mathbf{C}$ with the global features $C_{mn} = 1 - \text{sim}(\boldsymbol{z}^m, \boldsymbol{h}^n)$, where $\boldsymbol{z}^m$ and $\boldsymbol{h}^n$ denote the prompt-level features of $m$-th visual prompt and $n$-th textual prompt. However, the above definition primarily emphasizes prompt-level representation and might have a limited capacity to capture the detailed token-level features, *e.g.*, different patches within an image may capture various local region features. Thus, the obtained transport plan may fail to reflect the true relations between $P$ and $Q$. To this end, we further introduce the token-level OT that considers token-level alignments between two prompts. Specifically, we specify the visual output $\boldsymbol{x}$ and textual output $\boldsymbol{y}$ as two empirical distributions over token embeddings (here we omit the subscript $m$ and $n$ for clarity):

$$\boldsymbol{x} = \sum_{j=1}^{J} \frac{1}{J} \delta_{\boldsymbol{r}_j}, \qquad \boldsymbol{y} = \sum_{l=1}^{L} \frac{1}{L} \delta_{\hat{\boldsymbol{s}}_l},$$

where $\boldsymbol{r} = [\widetilde{\boldsymbol{e}}_1, ..., \widetilde{\boldsymbol{e}}_O, \widetilde{\boldsymbol{v}}_1, ..., \widetilde{\boldsymbol{v}}_b]$ is the output visual patches with length $J = b + O$, and $\boldsymbol{s} = [\widetilde{\boldsymbol{t}}_1, ..., \widetilde{\boldsymbol{t}}_b, \widetilde{\boldsymbol{w}}_{k,1}, ..., \widetilde{\boldsymbol{w}}_{k,k_l}]$ is the output textual tokens with length $b + k_l$. Unlike $\boldsymbol{z}$ and $\boldsymbol{h}$ that agent the prompt-level features, $\boldsymbol{x}$ and $\boldsymbol{y}$ collect the token-level features in the shared embedding space of CLIP. Naturally, the cost matrix $\hat{\mathbf{C}} \in R^{J \times L}$ in the token-level OT is defined as $\hat{C}_{jl} = 1 - \text{sim}(\boldsymbol{r}_j, \boldsymbol{s}_l)$, which measures the transport cost between the visual patches and textual tokens. As a result, the distance between $\boldsymbol{x}$ and $\boldsymbol{y}$ is the total transport cost of the token-level OT:

$$d_{\text{OT}}^{\lambda}(\boldsymbol{x}, \boldsymbol{y}; \hat{\mathbf{C}}) = d_{\text{OT}}^{\lambda}(\boldsymbol{x}, \boldsymbol{y}; \hat{\mathbf{C}}) - \lambda h(\hat{\mathbf{T}}), \tag{5}$$

where the transport plan $\hat{\mathbf{T}} \in R^{J \times L}$ denotes how likely is that the $j$-th visual patch transports to the $l$-th token feature, providing a principle solution to align token-level features. This motivated us to develop a combined cost matrix that considers prompt and token-level features together:

$$C_{mn} = 1 - \text{sim}(\boldsymbol{z}^m, \boldsymbol{h}^n) + \beta d_{\text{OT}}^{\lambda}(\boldsymbol{x}_m, \boldsymbol{y}_n; \hat{\mathbf{C}}^{mn}), \tag{6}$$

where $\beta$ is a trade-off parameter that controls the weight of token-level cost. The first two terms are the cosine distance between prompt-level features, and the last term is the OT distance between the token-level sets. In this way, Eq. 6 combines the pre-trained knowledge from two levels: the prompt-level features and the token-level embeddings. This enables the learned transport plan $\mathbf{T}$ in prompt-level OT to make fine-grained matching between $M$ visual and $N$ textual features, resulting in detailed alignments and better representations.

Once Eq. 4 is computed, we follow previous work [17] and predict the label of image $\mathbf{X}_j$ as:

$$p(y = k | \mathbf{X}_j) = \frac{\exp((1 - d_{\text{OT}}^{\lambda}(P_j, Q_k; \mathbf{C}^{jk})) / \tau)}{\sum_{k'=1}^{K} \exp((1 - d_{\text{OT}}^{\lambda}(P_j, Q_{k'}; \mathbf{C}^{jk'})) / \tau)}, \tag{7}$$

where $\mathbf{C}^{j,k}$ denote the cost matrix of $j$-th image and $k$-th label. Note that the weight of the classifier $Q_k$ in our model can be viewed as a discrete uniform distribution over $N$ textual prompts of label $k$, which contains multiple class-related semantics, improving the classification results. Thanks to the

differentiable Sinkhorn algorithm, all parameters of the proposed model can be optimized end-to-end by minimizing the following cross-entropy loss:

$$L = -\frac{1}{|\mathcal{X}|} \sum_{\mathbf{X} \in \mathcal{X}} \sum_{k=1}^{K} y_{\boldsymbol{x},c} p(y = k|\boldsymbol{x}). \tag{8}$$

where $y_{\mathbf{X}}$ is the one-hot label vector of image $\mathbf{X}$ Due to the OT formulation, our proposed `ALIGN` aims to learn $M$ visual prompt sequences and $N$ textual prompt sequences without introducing any neural networks. We describe our proposed model in the Appendix Algoritm. 1.

## 4   Related Work

**Single-modal prompt tuning:**   There are two storylines of single-modal prompt tuning, TPT and VPT. The former focuses on the language branch of a PLV and is interested in prompt learning in continuous embedding space. As one of the representative works, CoOp [3] models a prompt's context using a set of learnable vectors and shows great improvement over intensively-tuned manual prompts. To solve the weak generalizability on unseen category, CoCoOp [4] extends CoOp by explicitly conditioning prompts on image instances, which shifts the concentrations away from a specific set of classes to each input instance, enabling a stronger generalization performance. Instead of single-mode prompt learning, PLOT [17] learns multiple textual prompts by adopting the OT distance between prompts and image patches, achieving diverse prompt tuning. ProDA [19] first maturely designs multiple prompts and then models the uncertainty of prompts by employing the Gaussian distribution to model prompt embeddings. Correspondingly, VPTs refer to prepending visual patches to the image input space, which also shows impressive results in adapting PVLs into downstream tasks. For example, Jia et al. [6] introduces trainable visual prompt vectors into the image patch sequence of each Transformer layer and learns them along with a linear head. Despite the promising performance on various visual tasks, those models are designed to learn single-modal prompts, which fails to make use of the pre-trained multi-modal knowledge.

**Multi-modal prompt tuning:**   Moving beyond single-modal prompt tuning, MPT is a recently introduced task that learns textual prompts and visual prompts at the same time. This jointly tuning strategy not only distills the multi-modal knowledge but enables the dynamic alignments between prompts across modalities, showing better generalization. Zang et al. [27] propose a unified prompt tuning framework (UPT) [27] that shares an initial prompt across different modalities and designs a tiny network to generate the modality-specific prompts together. Almost parallel to UPT, Khattak et al. [28] proposed multi-modal prompt tuning (MaPLe) and adopted a projection matrix to condition vision prompts on their language counterparts explicitly allowing mutual propagation of gradients to promote synergy. In comparison, this work aims to learn multi-modal multi-mode prompts to better meet the requirement of diverse comprehensive representations. Besides, unlike measuring the similarity between images and labels by the global prompt-level features, we model each prompt as an empirical distribution over the token-level embedding space, and the similarity score is calculated by combining the prompt and token-level features under a hierarchical OT framework, which provides a novel and elegant tool to adapt PVLs into downstream tasks.

## 5   Experiments

### 5.1   Experimental Setup

**Datasets**   To make a comprehensive evaluation, we performed extensive experiments on 4 task settings, such as few-shot image recognition, base-to-new generalization, cross-dataset transfer learning, and domain generalization. Those experiments are conducted on 15 widely used image datasets, varying in scale and domains, including ImageNet [44], Caltech101 [45], OxfordPets [46], StanfordCars [47], Flowers102 [48], Food101 [49], FGVCAircraft [50], EuroSAT [51], UCF101 [52], DTD [53], SUN397 [54], ImageNetV2 [55], ImageNet-Sketch [56], ImageNet-A [57], and ImageNet-R [58]. The details of each dataset are provided in the Appendix Table. B. 1.

**Baselines**   We compare `ALIGN` with the state-of-the-art methods, including: CLIP [1], which provides the base results without prompt tuning; the single-modal prompt tuning methods,*e.g.*,

TPTs: CoOP [3], CoCoOp [4] and PLOT [17], and VPTs: VPT [6], and multi-modal prompt tuning methods: UPT [27] and MaPLe [28]. Note that we modified the official code of PLOT and changed the backbone to ViT-B/16 for a fair comparison.

**Implementation Details**    Following previous MaPLe [28], we load the pre-trained Vit-B/16 CLIP model as our backbone, where $d_l = 512$, $d_v = 768$ and $d = 512$. We set the number of textual and visual prompts $M = N = 4$, the length of prompt tokens $b = 2$, the hyperparameter $\lambda = 0.1$, and $\beta = 1$. The maximum iteration number in the Sinkhorn algorithm is set as 100. For all tasks, we train our model with a batch-size of 4, a learning rate of 0.0035, and an optimizer as SGD. For each task, we optimize the number of epochs. Following MaPLe we run 2 epochs to train ImageNet as a source model with a learning rate of 0.0026. The reported results are the averaged value over 3 seeds. Please refer to the Appendix Sec. B for more details. For all baselines, we set the length of prompts as 4 and collect their results according to the original papers or previous works. Thus, some experimental results may be missing.

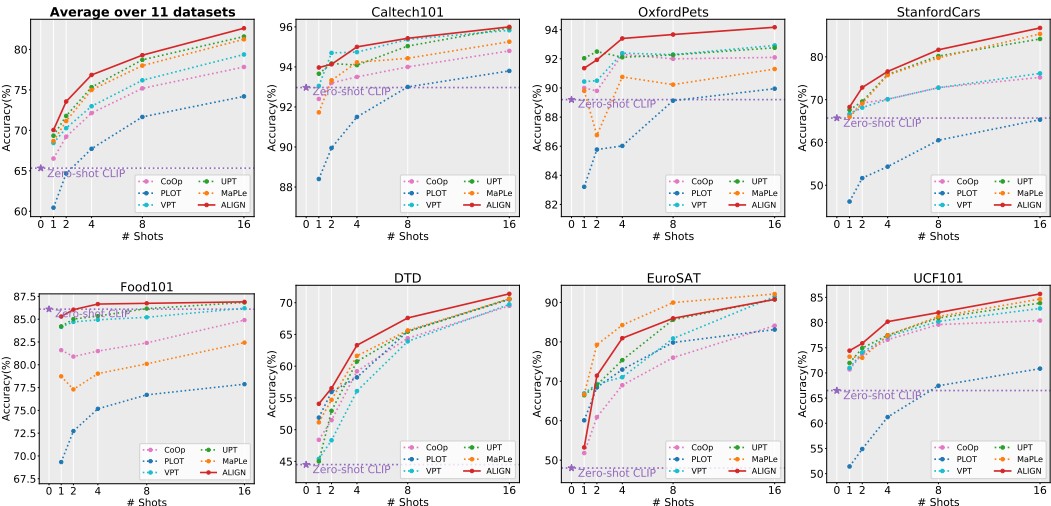

Figure 3: The few-shot learning results on 7 datasets (more detailed results of other datasets can be found in the Appendix Table. D. 1.). The red solid line denotes our ALIGN method, and the dotted lines represent various baselines. All results are reported as the mean value over three seeds.

### 5.2   Evaluation with the Standard Setting

**Few-Shot Learning.**    We first evaluate our model on few-shot classification, where models are trained on 1,2,4,8, and 16 shots and then applied to the test sets. We report the accuracy scores of all models across 11 datasets at Fig. 3. Overall, our proposed ALIGN outperforms others in most cases and achieves consistent improvement over CoOp on all datasets. Among the multi-modal prompt tuning methods, our method shows superior performance compared to UPV and MaPLe in general, except for the EuroSAT datasets. This demonstrates that ALIGN has the ability to distill the across-modalities knowledge and efficiently adapt PVLs into downstream tasks. Although there is a small margin between our model and those two models on some datasets, the competing models usually cannot achieve high performance over all datasets, *e.g.*, ALIGN exhibits 9.06/3.56/2.56/2.61/0.78(%) accuracy improvements compared with UPT on DTD datasets and achieves 6.56/8.75/7.63/6.64/4.47(%) improvements compared with MaPLe. In addition, we also find that our ALIGN performs better on 1/2/4 shots settings, showing the efficiency of our fine-grained alignments, which provide the token-level comparison during prediction. This capability contributes to more accurate classification, even with limited training samples.

**Base-to-New Generalization.**    To assess the generalizability of our model, we follow CoCoOp [4] to equally split the classes into base and new sets, and models are only trained on the base classes while tested on the new sets. Table. 1 reports the results, and we have the following observations: First, our proposed ALIGN surpasses previous baselines by achieving the highest average scores,

Table 1: Base-to-New on 11 datasets. The prompts are learned from the 16-shots base set. We report the classification accuracy on base set (Base), new set (New), and their harmonic mean (H), where H = $(2 \times \text{Base} \times \text{New})/(\text{Base} + \text{New})$. The best results are **highlighted**.

|  | Average | | | ImageNet | | | Caltech 101 | | | Oxford Pets | | |
|---|---|---|---|---|---|---|---|---|---|---|---|---|
|  | Base | New | H | Base | New | H | Base | New | H | Base | New | H |
| CLIP | 69.34 | 74.22 | 71.69 | 72.34 | 68.14 | 70.21 | 96.84 | 94.00 | 95.39 | 91.17 | 97.26 | 94.11 |
| CoOp | 82.66 | 63.22 | 71.65 | 76.14 | 67.88 | 71.77 | 98.00 | 89.81 | 93.72 | 93.67 | 95.29 | 94.47 |
| CoCoOp | 80.47 | 71.69 | 75.83 | 75.98 | 70.43 | 73.10 | 97.96 | 93.81 | 95.84 | 95.20 | 97.69 | 96.43 |
| PLOT | 77.20 | 60.38 | 67.76 | 75.97 | 69.23 | 72.44 | 96.53 | 82.86 | 89.17 | 93.45 | 79.76 | 86.06 |
| MaPLe | 82.28 | 75.14 | 78.55 | 76.66 | 70.54 | 73.47 | 97.74 | 94.36 | 96.02 | 95.43 | 97.76 | 96.58 |
| ALIGN | **83.38** | **75.51** | **79.25** | **76.89** | **72.15** | **74.45** | **98.37** | **94.70** | **96.50** | **95.67** | **97.93** | **96.79** |

|  | Stanford Cars | | | Flowers 102 | | | Food 101 | | | FGVC Aircraft | | |
|---|---|---|---|---|---|---|---|---|---|---|---|---|
|  | Base | New | H | Base | New | H | Base | New | H | Base | New | H |
| CLIP | 63.37 | 74.89 | 68.65 | 72.08 | **77.80** | 74.83 | 90.10 | 91.22 | 90.65 | 27.19 | 36.29 | 31.08 |
| CoOp | **78.12** | 60.40 | 68.12 | 97.60 | 59.67 | 74.06 | 88.33 | 82.26 | 85.18 | **40.44** | 22.30 | 28.74 |
| CoCoOp | 70.49 | 73.59 | 72.10 | 94.87 | 71.75 | 81.71 | 90.70 | 91.29 | 90.99 | 33.41 | 23.71 | 27.74 |
| PLOT | 61.41 | 42.69 | 50.37 | 95.26 | 56.03 | 70.56 | 88.45 | 85.28 | 86.84 | 29.63 | 16.17 | 20.92 |
| MaPLe | 72.94 | 74.00 | 73.47 | 95.92 | 72.46 | 82.56 | 90.71 | 92.05 | 91.38 | 37.44 | 35.61 | 36.50 |
| ALIGN | 77.24 | **76.38** | **76.80** | **97.70** | 73.3 | **83.75** | **90.77** | **92.07** | **91.42** | 37.56 | **36.97** | **37.26** |

|  | SUN 397 | | | DTD | | | EuroSAT | | | UCF 101 | | |
|---|---|---|---|---|---|---|---|---|---|---|---|---|
|  | Base | New | H | Base | New | H | Base | New | H | Base | New | H |
| CLIP | 69.36 | 75.35 | 72.23 | 53.24 | 59.90 | 56.37 | 56.48 | 64.05 | 60.02 | 70.53 | 77.50 | 73.85 |
| CoOp | 80.60 | 65.89 | 72.50 | 79.44 | 41.18 | 54.24 | 92.19 | 54.74 | 68.69 | **84.69** | 56.05 | 67.45 |
| CoCoOp | 79.74 | 76.86 | 78.27 | 77.01 | 56.00 | 64.85 | 87.49 | 60.04 | 71.21 | 82.33 | 73.45 | 77.64 |
| PLOT | 78.56 | 72.34 | 75.32 | 69.87 | 53.63 | 60.68 | 87.39 | 67.63 | 74.30 | 72.71 | 41.51 | 52.84 |
| MaPLe | 80.82 | 78.70 | 79.75 | 80.36 | **59.18** | **68.16** | **94.07** | 73.23 | 82.35 | 83.00 | 78.66 | 80.77 |
| ALIGN | **82.47** | **79.68** | **81.05** | 82.13 | 54.17 | 65.28 | 94.03 | **74.9** | 83.38 | 84.43 | **78.33** | **81.27** |

Table 2: Cross-dataset transfer learning accuracy results. Here we use the key letters to denote the datasets. The best results are **highlighted**.

| | Source | Target | | | | | | | | | | |
|---|---|---|---|---|---|---|---|---|---|---|---|---|
| Method | ImageNet | Cal | Pets | Cars | Flo | Food | FGCV | SUN397 | DTD | EuroSAT | UCF101 | Average |
| CoOp | 71.51 | 93.70 | 89.14 | 65.41 | 68.71 | 85.30 | 18.47 | 64.15 | 41.92 | 46.39 | 66.55 | 63.88 |
| CoCoOp | 71.02 | **94.43** | 90.14 | 65.32 | 71.88 | 86.06 | 22.94 | 67.36 | 45.73 | 45.37 | 68.21 | 65.74 |
| MaPLe | 70.72 | 93.53 | 90.49 | 65.57 | 72.23 | 86.20 | 24.74 | 67.01 | 46.49 | **48.06** | 68.69 | 66.30 |
| ALIGN | **72.03** | 93.91 | **90.55** | **65.84** | **73.75** | **86.40** | **24.95** | **67.59** | **46.75** | 47.25 | **69.60** | **67.15** |

thereby illustrating the superiority of the proposed framework. Second, ALIGN outperforms others in terms of H score across all datasets, except for the DTD dataset which indicates our method offers a more favorable trade-off between the base and new sets. We attribute this success to the token-level multi-mode prompt tuning strategy, where the multi-mode prompts enhance the ability to identify diverse visual concepts, which plays an essential role in unseen category prediction. Furthermore, for datasets that have small intra-class variances, such as Stanford Cars and FGVCAircraft, ALIGN achieves a noticeable improvement over MaPLe. The token-level alignment in ALIGN might account for this improvement, as it makes it more effective for fine-grained image classification.

**Transfer Learning and Domain Generalization.** To investigate the generalizability across-datasets or across-domains, we first train our model on ImageNet, utilizing all 1,000 classes, and subsequently apply it to 1) other 10 datasets and 2) other 4 domain shift datasets. We report those results at Table. 2 and 3, respectively. Based on those results, we find that our approach outperforms the baseline methods on 8/10 datasets with the best average accuracy score on dataset transfer learning task and 3/4 datasets on domain shift setting. These overall improvements highlight that ALIGN is less susceptible to the distribution shift between the source and target domains, thus revealing the robust generalizability of our model. Despite the marginal performance gain of ALIGN in contrast to MaPLe and UPT, our method outperforms them in most cases in terms of all four tasks and provides a novel multi-mode token-level alignment alternative for prompt tuning.

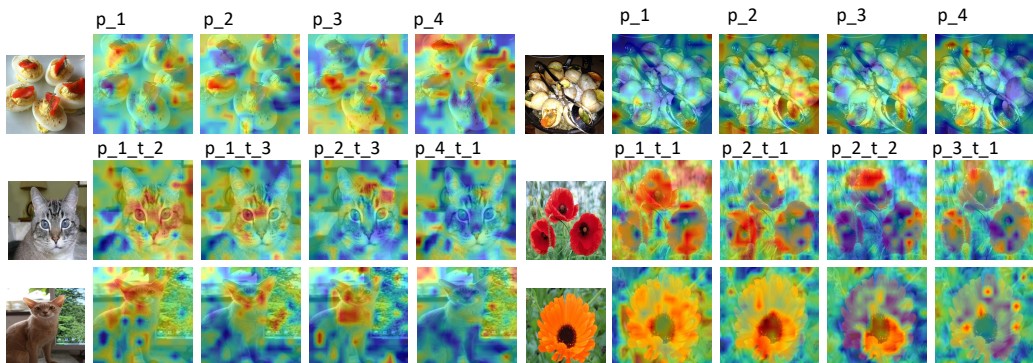

Figure 4: Visualization of the learned prompts and tokens. p_m denotes the $m$-th prompt and p_m_t_l denotes the $l$-th token of $m$-th prompt.

Table 3: Cross-domain generalization accuracy results. The best results are **highlighted**.

| Method | Learnable | Source | Target | | | |
| | | ImageNet | ImageNetV2 | ImageNet-Sketch | ImageNet-A | ImageNet-R |
|---|---|---|---|---|---|---|
| CLIP | ✗ | 66.73 | 60.83 | 46.15 | 47.77 | 73.96 |
| CoOp | ✓ | 71.51 | 64.20 | 47.99 | 49.71 | 75.21 |
| CoCoOp | ✓ | 71.02 | 64.07 | 48.75 | 50.63 | 76.18 |
| VPT | ✓ | 70.57 | 63.67 | 47.66 | 43.85 | 74.42 |
| UPT | ✓ | **72.63** | 64.35 | 48.66 | 50.66 | 76.24 |
| MaPLe | ✓ | 70.72 | 64.07 | 49.15 | 50.90 | **76.98** |
| ALIGN | ✓ | 72.03 | **64.64** | **49.96** | **50.94** | 76.16 |

**Qualitative Analysis** Besides the extensive quantitative results, we are also interested in the learned visual concepts of the proposed token-level alignments. Fortunately, the transport plan learned in our token-level OT provides us with access to a convenient tool to visualize the most related visual patches. We report the qualitative analysis at Fig. 2(b) and Fig. 4. From Fig. 2(b), we find that our model prefers to learn separable representations in both base and new classes. Recalling that $h^n$ denotes the global feature of $n$-th prompts, to visualize the learned prompt and obtain the attention map over the patch space, we calculate the cosine similarity between $h^n$ and all patch embeddings $\hat{e} \in R^{d \times O}$. We then view the normalized cosine similarity as the attention map and visualize the learned $N = 4$ prompts at the top row of Fig. 4. We observe that different prompts tend to align different patch regions, each of which contributes to the final prediction. This finding also meets with the motivation of the multi-mode prompt tuning, where each prompt aims to learn specific visual semantics.

Moving beyond the prompt-level visualization, we also visualize the token-level concepts. Specifically, for $l$-th column of the learned transport plan $\hat{T}_l \in R^J$ in token-level OT, it measures how likely the $l$-th token is transported to $J = b + O$ patches. Here we focus on the $O$ image patches and visualize the transport plan of an image sampled from the base set and new set at the second and third row in Fig. 4, respectively. We find that 1) different tokens within a prompt can capture various patches with similar visual concepts. For example, both p_1_t_2 and p_1_t_3 attend to the head of the cat; 2) Learning from the base set, the prompt tokens prefer to align the similar patches in the new set, which reveals that our token-level alignment module has the ability to transfer from the base set to the new set, rather than over-fitting to the base categories.

**Complexity Analysis** As discussed above, one of the key ideas of the proposed ALIGN is to learn multiple prompts for vision and language inputs and explore the token-level alignments under the hierarchical OT framework. To demonstrate the computation cost, we report the complexity analysis at Table. 4, where we focus on the number of trainable parameters (#Paras) and inference speed (fps). We find that 1) Overall, the multimodel prompts tuning methods (last three) require more trainable parameters and inference time than single-modal methods. 2) The proposed ALIGN requires

Table 4: Complexity analysis over various baselines. we report the number of trainable parameters (#Paras) and frames per second(fps). We can not report the fps result of UPT because of its unreleased code.

| Methods | CoOp | CoCoOp | VPT | PLOT | UPT | MAPLE | ALIGN |
|---------|------|--------|-----|------|-----|-------|-------|
| #Paras | 2,048 | 35,360 | 13,824 | 8,192 | 3,555,072 | 3,555,072 | 3,582,720 |
| fps | 645 | 37 | 152 | 583 | - | 282 | 62 |

slightly more training parameters than UPT and MAPLE because of the multiple prompts. And it also requires more inference time than MAPLE, due to the hierarchical OT operations. 3) Thanks to the independent OT operations, which can be calculated parallelly with the GPU, ALIGN has a faster testing time than CoCoOp, and achieves 62 fps at the test stage.

## 6 Conclusion

This paper introduces a novel multi-mode token-level alignment framework for multi-modal prompt tuning under optimal transport. We first employ the prompt-level OT to model the multi-mode prompts across modalities, and then introduce the token-level OT by viewing each prompt itself as a set over token embedding space. By coupling those two-level OT via the cost matrix, the final prediction is obtained by combining the prompt-level features and the token-level embeddings, enabling fine-grained alignments. Extensive experiments have been conducted, showing that our proposed model achieves competing performance on four settings. In terms of the **limitations**, the users may still need large GPU memory to load the pre-trained weights of PVLs to apply the proposed model to the test process. One potential solution is to combine prompt tuning with knowledge distillation. We leave it as a future study. Thanks to the open-world visual concept understanding of PVLs, our model shows promising zero-shot/few-shot ability for image recognition, which has the potential to encourage researchers to derive new and better methods for prompt tuning. Our work may indirectly lead to a negative **impacts** if there is a sufficiently malicious or ill-informed choice of a few-shot classification task.

## Acknowledgements

This work was supported in part by the National Natural Science Foundation of China under Grant U21B2006; in part by Shaanxi Youth Innovation Team Project; in part by the Fundamental Research Funds for the Central Universities QTZX23037 and QTZX22160; in part by the 111 Project under Grant B18039, in part by the Fundamental Research Funds for the Central Universities; in part by the Innovation Fund of Xidian University under Grant YJSJ23016;

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

**Appendix**

## A  Discussion

We in this paper propose ALIGN, a unified framework for multi-modal prompt tuning, where multi-mode modality-specific prompts are learned via the token-level alignment strategy. Moving beyond the single-model methods, which focus on textual prompt tuning or visual prompt tuning, ALIGN allows one to learn textual and visual prompts simultaneously, resulting in better representations in the shared vision-text embedding space. Compared to recent multi-modal methods, such as UPT [27] and MaPLe [28], ALIGN prefers to learn multi-mode prompts to capture diverse class attributes and develop the token-level alignment for fine-grained comparisons. This provides ALIGN with an efficient tool to calculate the similarity between prompts. We find that many previous works can be merged into our ALIGN framework with special hypermeter settings. We summarize this relationship at Table. 5. The N/A in Table. 5 means that PLOT calculates the similarity between the prompt-level label embeddings and the visual patch embeddings, which is not the case in ALIGN, where we calculate the similarity of prompt-level OT between textual label embeddings and visual image embeddings. and calculate the similarity of token-level OT between token embeddings and patch embeddings.

Table 5: Most previous works can be merged into our ALIGN framework. $M$: Number of visual prompts. $N$: Number of textual prompts. $\beta$: Weight of token-level OT in Eq.6 in the manuscript.

| Methods | Type | $M$ | $N$ | $\beta$ |
|---------|------|-----|-----|---------|
| CoOp [3] | Textal Prompt Tuning | 0 | 1 | 0 |
| VPT [6] | Visual Prompt Tuning | 1 | 0 | 0 |
| PLOT [17] | Textal Prompt Tuning | 0 | $\geq 1$ | N/A |
| UPT [27] | Multi-modal Prompt Tuning | 1 | 1 | 0 |
| MaPLe [28] | Multi-modal Prompt Tuning | 1 | 1 | 0 |
| ALIGN(Ours) | Multi-modal Prompt Tuning | $\geq 0$ | $\geq 0$ | $\geq 0$ |

## B  Data statistics and Hyperparameter setting

We thoroughly evaluate our proposed ALIGN framework across four distinct tasks: few-shot recognition, base-to-new generalization, cross-dataset transfer, and cross-domain generalization. These extensive experiments are conducted on a diverse set of fifty commonly used vision datasets, covering various contexts. These datasets include ImageNet [44] and Caltech101 [45] for generic image classification, OxfordPets [46], StanfordCars [47], Flowers102 [48], Food101 [49], and FGVCAircraft [50] for fine-grained image recognition, SUN397 [54] for scene recognition, UCF101 [52] for action recognition, DTD [53] for texture classification, and EuroSAT [51] for satellite imagery recognition. In the case of the cross-domain generalization task, our model is trained on ImageNet and subsequently tested on ImageNetV2 [55], ImageNet-Sketch [56], ImageNet-A [57], and ImageNet-R [58]. We summarize data statistics at Table. B. 1

The evaluation pipeline for each task follows the approach employed by previous works [3, 28]. The specific details of this pipeline are summarized below:

**Few-shot Recognition.**  To evaluate the efficiency of the proposed ALIGN on the few-shot case, we follow CoOp [3], and first partition the dataset into base and novel sets. Those two sets share the same categories. Models are trained on the base set using a variety of shot settings, including 1, 2, 4, 8, and 16 shots per class, and then tested on the full novel set. The accuracy scores are reported to compare the performance. The training epoch is set as 10 for 1, 2, and 4 shots and 40 for 8 and 16 shots.

**Base-to-New Generalization.**  To show the Generalizability of unseen categories, we first divide the dataset into two separate subsets: the base subset and the new subset. Importantly, these subsets do not share the same categories. The base subset contains a specific set of categories used for model training, while the new subset consists of previously unseen categories that the model has not

been exposed to during training. Besides reporting the accuracy score on base and novel sets, we also calculate the harmonic mean $H = (2 \times \text{Base} \times \text{New})/(\text{Base} + \text{New})$, which acts as a trade-off between Base and New, providing a comprehensive measure of overall model performance. The training epoch is set as 8.

**Cross-Dataset Transfer.**    To determine the transferability of our model across different datasets, we first train our model on the source dataset (ImageNet) and then evaluate it on 10 different target datasets. The training epoch is set as 2 and the learning rate is set as 0.0026.

**Cross-Domain Generalization.**    To determine the robustness of our model on the distribution-shift setting, we trained our model on the source dataset (ImageNet) and then assess it on 4 domain-shifted datasets, including ImageNetV2, ImageNet-Sketch, ImageNet-A, and ImageNet-R. The training epoch is set as 2 and the learning rate is set as 0.0026.

The other training hyperparameters in the previous experiments are set according to MaPLe [28], which are detailed listed at Table B. 2.

Table B. 1: Statistics of the used 15 datasets. N/A denotes that we do not use the corresponding training or validation sets.

| Dataset | Domains | #Classes | #Train | #Val | #Test |
|---|---|---|---|---|---|
| ImageNet | generic object | 1000 | 1.28M | N/A | 50,000 |
| Caltech101 | generic object | 100 | 4,128 | 1,649 | 2,465 |
| OxfordPets | fine-grained object | 37 | 2,944 | 736 | 3,669 |
| StanfordCars | fine-grained object | 196 | 6,509 | 1,635 | 8,041 |
| Flowers102 | fine-grained object | 102 | 4,093 | 1,633 | 2,463 |
| Food101 | fine-grained object | 101 | 50,500 | 20,200 | 30,300 |
| FDVCAircraft | fine-grained object | 100 | 3,334 | 3,333 | 3,333 |
| SUN397 | scene recognition | 397 | 15,880 | 3,970 | 19,850 |
| UCF101 | action recognition | 101 | 7,639 | 1,808 | 3,783 |
| DTD | texture recognition | 47 | 2,820 | 1,128 | 1,692 |
| EuroSAT | satellite object | 10 | 13,500 | 5,400 | 8,100 |
| ImageNetV2 | generic object | 1000 | N/A | N/A | 10,000 |
| ImageNet-Sketch | sketch object | 1000 | N/A | N/A | 50,889 |
| ImageNet-A | generic object | 200 | N/A | N/A | 7,500 |
| ImageNet-R | generic object | 200 | N/A | N/A | 30,000 |

## C    Training Algorithm

Given the training datasets $\mathcal{D} = \{\mathcal{X}_i, y_{\mathcal{X}_i}\}_{i=1}^{N_\mathcal{D}}$, our method aims to learn $M$ visual and $N$ textual prompts simultaneously. All parameters in ALIGN are optimized by minimizing the cross-entropy loss end-to-end. We summarize the training algorithm at Algorithm. 1.

Table B. 2: Hyperparameter setting used in the previous experiments.

| Hyperparameters | Values |
|---|---|
| Batch Size | 4 |
| Input Size | $224 \times 224$ |
| Input Interpolation | "Bicubic" |
| Input Pixel Mean | $[0.48145466, 0.4578275, 0.40821073]$ |
| Input Pixel STD | $[0.26862954, 0.26130258, 0.27577711]$ |
| Transforms | ["random resized crop", "random filp", "normalize"] |
| Optimizer | SGD |
| Learning Rate | 0.0035 |
| LR Scheduler | "cosine" |
| Warmup Epoch | 1 |
| Warmup Type | "constant" |
| Warmup LR | $1e$-5 |
| Backbone | ViT-B/16 |
| Number of Textual Prompts | 4 |
| Number of Visual Prompts | 4 |
| Learnable Prompt Length | 2 |
| Fixed Prompt Length | 2 |
| weight of token-level cost | 1 |
| weight of regularization in OT | 0.1 |
| Prompt Initialization | "a photo of a" |
| Precision | "fp16" |

---

**Algorithm 1** Training algorithm of ALIGN.

---

**Input**: Training dataset $\mathcal{D}$, a pre-trained vision-language model, class name set, number of visual prompts $M$, number of textual prompts $N$, and the training epoch.
**Output**: The learned ALIGN, which discovers multi-modal multi-mode prompts for downstream tasks.
**Initialize**: The $M$ and $N$ multi-modal prompt embeddings.
**Preprocess**: Built $N \times K$ textual token inputs according to Sec 2.1 in the manuscript.
**for** iter = 1,2,3,... **do**
    **1.** Feed the textual input into the text encoder $g$ and collect the outputs with the corresponding prompt-level representation $\{h_k^n\}_{k=1,n=1}^{K,N}$ and token embeddings $\{s_k^n\}_{k=1,n=1}^{K,N}$, where each $s_k^n$ is the output token embeddings of $n$-th prompt of $k$-th label with length $b + k_l$.
    **2.** Sample a batch of $J$ images. Built $N \times B$ visual patch inputs according to Sec2.1 in the manuscript. Feed the visual input into the visual encoder $f$ and collect the outputs with the corresponding prompt-level representation $\{z_j^m\}_{j=1,m=1}^{J,M}$ and patch embeddings $\{r_j^m\}_{j=1,m=1}^{J,M}$, where each $r_j^m$ denotes the output patch embeddings of $m$-th prompt of $j$-th image with length $b + O$.
    # Two-level OT
    **3.** Calculate the token-level OT distance between each image and each label in Eq.5 with the collected patch set and token set.
    **4.** Calculate the cost matrix in prompt-level OT according to Eq.6, and then get the prompt-level OT distance in Eq.4.
    Compute the cross-entropy loss $L$ with the obtained prompt-level OT distance according to Eq.8 and update all learnable parameters by minimizing $L$ with the stochastic gradient descent algorithm.
**end for**

---

# D Additional Results

We in this section report additional results of other datasets on the few-shot task and conduct the ablation studies on the prompt and token-level OT.

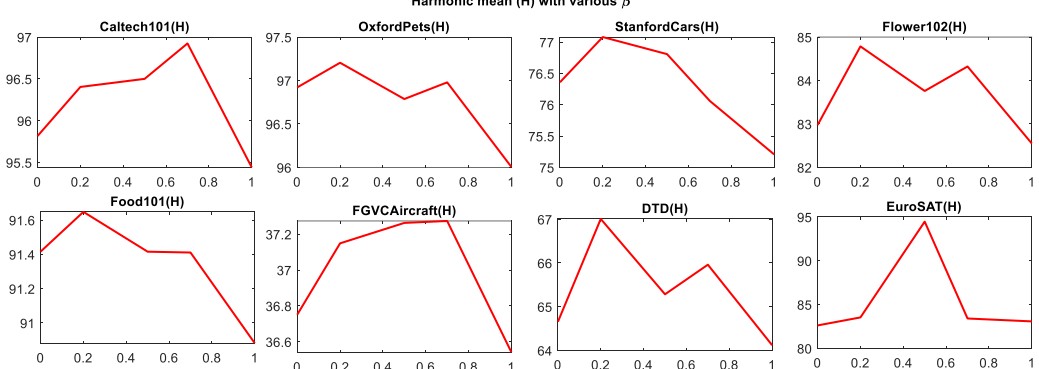

Figure D1: Harmonic mean (H) results of ALIGN on Base-to-New task under different $\beta$.

## D.1 Few-shot Results

We report the numerical results of various methods on 11 datasets at Table. D. 1.. From the results, we find that our method ALIGN outperforms baselines in most cases, which demonstrates the efficiency of the token-level prompt alignment module.

## D.2 Ablation studies

Recall that the proposed model consists of the prompt-level and token-level OT, which align the textual and visual modalities from hierarchical semantics. In the previous experiment, we view the prompt-level and token-level OT equally and set the hyperparameter weight $\beta = 1$ in Eq.6 in the manuscript. Here want to analyze how those two OTs affect the model performance. To this end, we rewrite the cost matrix in Eq.6 in the manuscript as:

$$C_{mn} = (1 - \beta)(1 - \text{sim}(\boldsymbol{z}^m, \boldsymbol{h}^n)) + \beta d_{\text{OT}}^{\lambda}(\boldsymbol{x}_m, \boldsymbol{y}_n; \hat{\mathbf{C}}^{mn}). \qquad (9)$$

Note that $\beta = 0$ and $\beta = 1$ denote two of our variants, where the former denotes only prompt-OT works and the latter means we only focus on token-level similarity. We report the ablation results of ALIGN on Base-to-New tasks under various settings, *e.g.*, $\beta = [0, 0.2, 0.5, 0.7, 1.0]$ at Fig. D1. We have the following interesting findings: 1) The combined ALIGN works better than each of them; 2) After finetuning $\beta$ for each dataset, one can obtain better results than the reported values in our paper.

Table D. 1.: The few-shot results of various methods on 11 datasets. We report mean value over 3 different seeds. The best results are **highlighted**.

| Dataset | Methods | 1 shot | 2 shots | 4 shots | 8 shots | 16 shots |
|---|---|---|---|---|---|---|
| Caltech101 | CoOp | 92.4 | 93.2 | 93.5 | 94.0 | 94.8 |
| | PLOT | 88.40 | 89.95 | 91.50 | 93.00 | 93.24 |
| | UPT | 93.66 | 94.17 | 94.09 | 95.04 | 95.95 |
| | MaPLe | 91.73 | 93.33 | 94.23 | 94.43 | 95.26 |
| | ALIGN | **93.97** | **94.13** | **95.00** | **95.43** | **96.00** |
| DTD | CoOp | 48.4 | 51.5 | 59.2 | 64.4 | 69.5 |
| | PLOT | 51.90 | 55.95 | 58.24 | 65.50 | 70.52 |
| | UPT | 45.01 | 52.97 | 60.74 | 65.44 | 70.62 |
| | MaPLe | 51.16 | 54.70 | 61.63 | 65.63 | 70.60 |
| | ALIGN | **54.07** | **56.53** | **63.3** | **67.6** | **71.40** |
| EuroSAT | CoOp | 51.8 | 60.9 | 69.0 | 76.0 | 84.1 |
| | PLOT | 60.10 | 68.45 | 72.97 | 79.84 | 83.12 |
| | UPT | 66.46 | 69.07 | 75.36 | 85.62 | 90.77 |
| | MaPLe | **66.67** | **79.26** | **84.25** | **89.96** | **92.14** |
| | ALIGN | 53.23 | 71.43 | 80.93 | 85.97 | 90.77 |
| FGVCAircraft | CoOp | 24.2 | 25.8 | 27.9 | 32.7 | 34.2 |
| | PLOT | 21.50 | 21.71 | 23.96 | 27.02 | 30.24 |
| | UPT | 28.43 | 29.91 | 33.34 | 39.50 | 46.61 |
| | MaPLe | 26.64 | 27.86 | 33.56 | 40.66 | 49.93 |
| | ALIGN | **29.57** | **31.63** | **34.03** | **40.95** | **49.99** |
| Flowers102 | CoOp | 72.9 | 80.4 | 85.7 | 92.3 | 96.2 |
| | PLOT | 70.00 | 81.34 | 88.29 | 92.84 | 95.10 |
| | UPT | 74.97 | 81.81 | 91.90 | 95.17 | **97.41** |
| | MaPLe | 80.24 | 88.14 | 90.07 | 95.10 | 96.34 |
| | ALIGN | **81.33** | **88.77** | **92.53** | **95.43** | 96.57 |
| FOOD101 | CoOp | 81.6 | 80.9 | 81.5 | 82.4 | 84.9 |
| | PLOT | 69.10 | 72.89 | 74.89 | 76.70 | 77.87 |
| | UPT | 84.21 | 85.01 | 85.34 | 86.16 | 86.83 |
| | MaPLe | 78.73 | 77.30 | 79.03 | 80.10 | 82.43 |
| | ALIGN | **85.29** | **86.05** | **86.66** | **86.74** | **86.90** |
| ImageNet | CoOp | 68.07 | 69.26 | 69.60 | 70.35 | 71.53 |
| | PLOT | 67.51 | 68.80 | 70.00 | 70.21 | 71.40 |
| | UPT | 69.55 | 69.88 | 70.28 | 71.58 | 72.64 |
| | MaPLe | 69.56 | 69.94 | 70.65 | **71.80** | **72.74** |
| | ALIGN | **69.80** | **70.02** | **70.84** | 71.77 | 72.45 |
| OxfordPets | CoOp | 90.0 | 89.8 | 92.3 | 92.0 | 92.1 |
| | PLOT | 83.21 | 85.77 | 86.02 | 89.13 | 89.95 |
| | UPT | 82.93 | 85.40 | 85.97 | 87.40 | 88.10 |
| | MaPLe | 89.80 | 86.76 | 90.76 | 90.23 | 91.30 |
| | ALIGN | **91.36** | **91.93** | **93.4** | **93.67** | **94.17** |
| StanfordCars | CoOp | 66.4 | 69.2 | 70.1 | 72.8 | 75.2 |
| | PLOT | 46.20 | 51.67 | 54.35 | 60.52 | 65.32 |
| | UPT | 67.60 | 69.57 | 75.88 | 80.19 | 84.17 |
| | MaPLe | 65.96 | 69.10 | 75.73 | 79.76 | 85.36 |
| | ALIGN | **68.27** | **72.84** | **76.58** | **81.65** | **86.75** |
| SUN397 | CoOp | 65.2 | 66.6 | 68.1 | 70.5 | 73.2 |
| | PLOT | 55.33 | 60.02 | 63.21 | 66.02 | 67.98 |
| | UPT | 68.84 | 69.76 | **72.12** | 74.00 | 75.90 |
| | MaPLe | 61.73 | 63.23 | 67.60 | 69.13 | 73.00 |
| | ALIGN | **69.14** | **69.98** | 71.88 | **74.15** | **76.57** |
| UCF101 | CoOp | 70.7 | 73.8 | 76.6 | 79.6 | 80.4 |
| | PLOT | 51.42 | 54.89 | 61.23 | 67.45 | 70.85 |
| | UPT | 71.98 | 74.93 | 77.49 | 80.91 | 83.86 |
| | MaPLe | 73.23 | 73.00 | 77.45 | 81.2 | 84.67 |
| | ALIGN | **74.42** | **75.87** | **80.18** | **81.99** | **85.69** |

