# OpenReview forum: "Tuning Multi-mode Token-level Prompt Alignment across Modalities"
_NeurIPS.cc/2023/Conference — NeurIPS 2023 poster_

### Official Review · Reviewer_dk2b · 2023-07-04

**Soundness:** 2 fair
**Presentation:** 2 fair
**Contribution:** 2 fair
**Rating:** 4
**Confidence:** 3

**Summary:**

The paper aims at improving few-shot classification based on CLIP-ViT-B/16 using prompt tuning. The authors introduced a multi-modal prompt tuning framework with token-level alignment and distribution matching. The prompts are tuned on ImageNet and then the model is benchmarked on 15 image datasets.

**Strengths:**

+ The method achieves better results than baselines (e.g., CoOp, MaPLe) on a series of datasets, including both few-shot classification and transfer learning tasks.
+ The code is attached, making the method reproducible.
+ The introduced approach should be practical in real-world applications.

**Weaknesses:**

- The writing appears rushed and some sections are not easy to follow. There are also some typos and grammar issues, e.g., L135-136 "...CLIP. maximuming...".
- The central insight of this work is to introduce token-level alignment in a multimodal prompt-tuning framework. However, the fine-grained alignment with optimal transportation is not new (e.g., Hierarchical Optimal Transport for Multimodal Distribution Alignment, NeurIPS19) and the multimodal prompt-tuning framework is similar to existing works like MPT. It provides limited new knowledge to the community.
- In the experimental part, there are only comparisons based on base-size models. However, it is pretty essential to verify the scalability of the introduced method on larger models.

**Questions:**

Please see the Weaknesses.

**Limitations:**

The authors have discussed their limitations on the GPU memory during test. It may also be important to discuss their training cost.

---

> ### Author Rebuttal · Authors · 2023-08-10
>
> We thank reviewer dk2b for the comments and suggestions. Below, we address the concerns raised in your review point by point. Please let us know if you have any further concerns or whether this adequately addresses all the issues that you raised with the paper.
>
> > The writing appears rushed and some sections are not easy to follow. There are also some typos and grammar issues, e.g., L135-136 "...CLIP. maximuming...".
>
> We will carefully improve the writing, reorganize the sections, and correct typos in the revision.
>
> > The central insight of this work is to introduce token-level alignment in a multimodal prompt-tuning framework. However, the fine-grained alignment with optimal transportation is not new (e.g., Hierarchical Optimal Transport for Multimodal Distribution Alignment, NeurIPS19) and the multimodal prompt-tuning framework is similar to existing works like MPT. It provides limited new knowledge to the community.
>
> First, we would like to note that one of the main contributions of this paper is to develop a unified multi-model prompt tuning method under the HOT framework, e.g., most previous works can be viewed as a particular case of our framework (Table 1 in the Appendix).
>
> Second, We agree that the token-level alignment module shares a similar idea with the previous Hierarchical Wasserstein Alignment algorithm Technically. However, this does not imply that the proposed methodology is not innovative. Our model is different from the previous work in terms of both tasks and objective functions. The proposed ALIGN views the transport distance as the similarity of prompts and the model is optimized by the classification loss. While the previous work focused on aligning clustered datasets and trained the model by minimizing the transport distance unsupervised. It is not a trivial case that directly employs the previous method on the prompt tuning task.
>
> Third, compared to existing MPTs, which often learn single modality-specific prompt sentence, our method provides the community with a novel alternative that prefers to learn multiple prompts, resulting in diverse concept discovery. Mathematically, we formulate the prompt tuning task as a HOT problem, which distinguishes the proposed model from previous MPTs.
>
> Last, the transport plan provides users with a visualization tool to explain the learned prompts, while the previous MPTs fail to give such interpretability.
>
>
>
>
> > In the experimental part, there are only comparisons based on base-size models. However, it is pretty essential to verify the scalability of the introduced method on larger models.
>
> Technically, the proposed method can be easily applied to pre-trained two-tower models. In the manuscript, we follow a series of previous works and load the pre-trained CLIP as our backbone. Extensive experiments are conducted to evaluate the superiority of the proposed model. We believe that those empirical results support this paper. We would appreciate it if the reviewers could provide some references that apply prompt tuning on large-scale vision-language models.
>
>
>
> > The authors have discussed their limitations on the GPU memory during test. It may also be important to discuss their training cost.
>
> We have reported the detailed comparison in the global comment section. Please check for more discussion.

---

> > ### Author Response · Authors · 2023-08-18
> > **Following up with Reviewer dk2b**
> >
> > Dear Reviewer dk2b,
> >
> > Thanks again for your effort in reviewing our paper and give us a great chance to improve the paper quality. We hope that our response can address your concerns.
> >
> > Considering that the discussion period will end on Aug 21st, we would like to know if you have any other questions about our paper, and we are glad to have a discussion with you in the following days. If our response has addressed your concerns, would you mind considering re-evaluating our work based on the updated information?
> >
> > Best regards,
> >
> > Authors

---

> > ### Comment · Reviewer_dk2b · 2023-08-19
> > **Response**
> >
> > Thanks for the authors' response, which addressed some of my concerns, e.g., the differences from existing works. Unfortunately, the authors did not report the results on larger sizes of CLIP models, which raises concerns that the proposed method may not work on large models. There's a case that, since larger sizes of foundation models are strong enough, such advanced prompt alignment method fails to yield extra gains compared to existing prompt alignment methods. So I choose to maintain my initial rating.

---

> > > ### Author Response · Authors · 2023-08-20
> > > **Additional results on larger size ViT-H/14 CLIP**
> > >
> > > We thank Reviewer dk2b for your replies and for specifying the main concern. Like previous works, we conducted extensive experiments by loading the ViT-B/16 as the CLIP model. As clarified in the previous rebuttal, technically, our method can be easily applied to CLIP-like modes and improve their performance. Following the reviewer's suggestion, we have evaluated our method based on ViT-H/14, which consists of a 32-layer image encoder and a 24-layer text encoder. The results of Base-to-New and Few-shot tasks are listed below (results are reported as the mean value of three seeds).
> > >
> > >                                       Base-to-New results (Base New H)
> > >
> > > | Methods |  | Cal |  |  |  |  |  | DTD |  |  |  |  |  | Eur |  |
> > > |:---:|:---:|:---:|:---:|:---:|:---:|:---:|:---:|:---:|:---:|:---:|:---:|:---:|:---:|:---:|:---:|
> > > | CLIP | 98.4 | 95.4 | 96.8 |  |  |  | 74.5 | 70.5 | 72.4 |  |  |  | 73.0 | 83.7 | 77.9 |
> > > | MaPLe | 98.0 | 94.9 | 96.4 |  |  |  | 83.2 | 74.4 | 78.5 |  |  |  | 96.2 | 84.5 | 89.9 |
> > > | ALIGN | 99.2 | 95.7 | 97.4 |  |  |  | 85.8 | 75.7 | 80.4 |  |  |  | 96.6 | 84.9 | 90.3 |
> > >
> > > | Methods |  | Pets |  |  |  |  |  | Cars |  |  |  |  |  | UCF |  |
> > > |:---:|:---:|:---:|:---:|:---:|:---:|:---:|:---:|:---:|:---:|:---:|:---:|:---:|:---:|:---:|:---:|
> > > | CLIP | 96.4 | 98.9 | 97.6 |  |  |  | 91.2 | 97.1 | 94.0 |  |  |  | 82.5 | 83.0 | 82.7 |
> > > | MaPLe | 96.7 | 98.3 | 97.4 |  |  |  | 90.3 | 96.8 | 93.4 |  |  |  | 86.3 | 84.2 | 85.2 |
> > > | ALIGN | 97.0 | 98.8 | 97.8 |  |  |  | 91.6 | 97.3 | 94.3 |  |  |  | 86.3 | 84.5 | 85.4 |
> > >
> > >
> > >                                       Few-shot results (1/2/4/8 shots)
> > >
> > > | Methods |  | Eur |  |  |  |  |  |  | Pets |  |  |
> > > |:---:|:---:|:---:|:---:|:---:|:---:|:---:|:---:|:---:|:---:|:---:|:---:|
> > > | CLIP | 67.0 | 67.0 | 67.0 | 67.0 |  |  |  | 94.5 | 94.5 | 94.5 | 94.5 |
> > > | MaPLe | 73.5 | 75.2 | 79.7 | 85.1 |  |  |  | 92.1 | 92.7 | 93.8 | 94.0 |
> > > | ALIGN | 77.3 | 78.1 | 80.0 | 89.5 |  |  |  | 93.9 | 94.5 | 94.9 | 94.8 |
> > >
> > > Due to the limited time, we report the Base-to-New results on Caltech 101, DTD, EuroSAT, Oxford Pets, Stanford Cars and UCF 101 datasets, and report the Few-shot results on EuroSAT and Oxford Pets datasets.
> > >
> > > From the results, we find that 1), Thanks to the larger size ViT-H/14, the performances of all three methods are improved with a large gap; 2) Our method outperforms the CLIP and MaPLe in most cases, this demonstrates the robustness of our proposed ALIGN method over backbone networks of different sizes. We will add these results in our revision.
> > >
> > > Thank you again for the valuable suggestion, which led to a more solid submission. We hope the above results can address your concern well. We are glad to have further discussion with you. Please feel free to contact us if you have any questions.

---

### Official Review · Reviewer_Uuq5 · 2023-07-04

**Soundness:** 3 good
**Presentation:** 3 good
**Contribution:** 3 good
**Rating:** 6
**Confidence:** 4

**Summary:**

This work aims to overcome the limitations of previous works in prompt tuning for vision-language models. Unlike prior approaches that focus on single modality or holistic prompt alignment, the paper proposes a multi-mode token-level tuning framework that leverages optimal transportation to align prompt tokens across different modalities. The framework introduced in the paper relies on two crucial elements: multi-mode prompt discovery and token-level alignment. By enabling diverse semantic representations, multi-mode prompt discovery ensures a broader range of prompts. On the other hand, token-level alignment allows for a more detailed exploration of similarity between modalities. Extensive experiments demonstrate the effectiveness of the new method ALIGN.

**Strengths:**

1. Overall, this work is well-motivated, and the paper is well-written. It focuses on a practical problem in prompt learning and proposes a novel method called ALIGN.
2. This work first leverages Optimal Transportation to address the limitations of prompt tuning. Intuitively, the new method can succeed in finding out token level vision-language prompts. Compared with the single-mode or holistic level prompt tuning approaches, this method can better reveal the connection between visual and textual prompts, which would be a very valuable insight for the study of vision-language alignment.
3. The authors conduct extensive experiments to evaluate their method, and the results demonstrate non-trivial improvements over the existing baseline models. The comparison is clear, and their method ALIGN achieves SoTA performance in most scenarios.


**Weaknesses:**

1. The paper lacks discussion of computing cost. It seems that your ALIGN requires more self-attention computation compared to the baselines such as VPT and TPT. Your improvements might be challenged if the difference in training/inference cost is not provided or is too big.
2. While the paper demonstrates its merits in token-level prompt learning, more empirical results in fine-grained tasks such as semantic segmentation should be included. However, the paper only gives classification results. I understand that implementing prompts into segmentation tasks is not easy and it merely appears in your baselines, but it would be a very strong support to your ALIGN’s effectiveness and significance.
3. Some expressions are confusing. For example, in line 249-250, there is “For each task, we optimize the number of epochs”. Do you mean “the same number of epochs”?


**Questions:**

1. How about the comparison of computing cost between your method and the baselines?
2. Is there any experimental results in inference tasks other than classification?

**Limitations:**

See "Weaknesses".

---

> ### Author Rebuttal · Authors · 2023-08-10
>
> We thank reviewer Uuq5 for providing positive feedback and helpful suggestions. Below are our responses.
>
>
> > Lacks discussion of computing cost and Q1: How about the comparison of computing cost between your method and the baselines?
>
> We have reported the detailed comparison in the global comment section. Please check for more discussion.
>
> > Is there any experimental results in inference tasks other than classification? and more empirical results in fine-grained tasks such as semantic segmentation should be included and Q2: Is there any experimental results in inference tasks other than classification.
>
> We thank the reviewer for pointing out the potential applications of the proposed model. The proposed model proposed a novel token-level alignment for multi-modal prompt tuning tasks. It improves the classification results by aligning the semantics of the visual patches and the textual tokens. Following the previous empirical setting, we conduct extensive experiments on 4 classification tasks on 12 datasets. We appreciate the reviewer's suggestion and will leave the semantic segmentation to future work. Also, we would be very happy if the reviewer could suggest relevant papers.
>
> > For each task, we optimize the number of epochs.
>
> Following previous work, we use different epochs for 4 tasks. We will correct the statement in the revision.

---

> > ### Comment · Reviewer_Uuq5 · 2023-08-20
> > **Response to authors' rebuttal**
> >
> > Thank you for your rebuttal. My concerns are addressed, and I keep my rating of weak acceptance.

---

> > > ### Author Response · Authors · 2023-08-21
> > >
> > > Thank you for the replies, We are glad that our response address your concerns. We will revise our work accordingly.

---

### Official Review · Reviewer_hEEC · 2023-07-07

**Soundness:** 2 fair
**Presentation:** 3 good
**Contribution:** 3 good
**Rating:** 6
**Confidence:** 4

**Summary:**

This paper introduces a multi-mode token-level alignment framework for multi-modal prompt tuning, which improves the representation of visual and textual modalities and can be used to improve existing methods. The task is formulated as a distribution matching problem, addressed using prompt and token-level optimal transportation (OT), providing a principled and elegant solution. The method is applied to few-shot classification, dataset transfer learning, and domain generalization, showing superior results on widely used datasets.

**Strengths:**

• The learning of multi-modal, multi-mode prompts is facilitated by establishing optimal transport (OT) at the prompt and token level.
• The structure of the manuscript is solid and it's well-written overall.
• The efficiency of the proposed ALIGN method for both few-shot classification and generalization has been confirmed through a series of diverse experiments.

**Weaknesses:**

• The proposed model might be memory-intensive, however, an analysis of the additional time and memory costs has not been provided.
• The omission of specific details, particularly regarding the ablation analysis, somewhat undermines the impressive results.
  - The study does not examine the influence of prompt length and quantity on the experimental outcomes.
  - It remains unclear whether token-level alignment provides any enhancements compared to prompt-level alignment.

• Miscellaneous issue
- Figure 1 is not mentioned in the body text.
- It seems that line 135 is missing a period, and the expression "maximuming" appears to be awkward.
- In Table 1, CoOp shows the best result in 'Stanford Cars'-Base, so it should be highlighted instead of ALIGN

**Questions:**

See the comments in weakness.

**Limitations:**

• As pointed out in the conclusion section, this paper's method still demands substantial GPU memory.
• The method suggested does not appear to be well-suited for a fully zero-shot scenario where there is no training samples. This scenario is little bit different from the Base-to-New Generalization scenario.

---

> ### Author Rebuttal · Authors · 2023-08-10
>
> We thank reviewer hEEC for providing positive feedback and helpful suggestions. Below are our responses.
>
> > Analysis of the additional time and memory costs has not been provided.
>
> We have reported the detailed comparison in the global comment section. Please check for more discussion.
>
> > Additional results of prompt length and quantity.
>
> | Length | 2  | 4| 8 |16 |
> |:--:|:----:|:--:|:--:|:--:|
> |  UCF 101 | 81.27 | 81.24 | 81.45 | 81.42 |
> |  Flowers102| 83.75 | 83.84 | 83.32 | 83.57 |
> | Stanford Cars | 76.80 | 76.68 | 77.14 | 76.82 |
>
>
> |Number | 1 | 2  |  4  |   8   |
> |:--:|:-----:|:--:|:--:|:--:|
> | UCF 101| 80.84 | 81.13 | 81.27 | 81.42 |
> | Flowers102| 82.64 | 82.97 | 83.75 | 83.43 |
> | Stanford Cars | 74.18 | 75.32 | 76.80 | 76.73 |
>
> Following your advice, we have reported the results of prompt length and quantity. We fix the prompt length as 2 and set the number of prompts as 4 according to the previous works in the manuscript. From the added results, we find that the proposed model enjoys good robustness of prompt length and quantity. And one may have better results than we report when finetuning those hyperparameters.
>
>
> > Unclear whether token-level alignment provides any enhancements compared to prompt-level alignment.
>
> |  Methods  | UCF 101 | Stanford Cars | Flowers102 | Oxford Pets |
> |:--:|:---:|:--:|:----:|:---:|
> | ALIGN w/o prompt |  80.84  | 74.18 |  82.64 | 96.51  |
> |  ALIGN w/o token |  81.04  |  75.32  |  83.09 | 96.61  |
> |   ALIGN   |  81.27  |  76.80  |  83.75   | 96.79 |
>
> We have reported the ablation results of the introduced two modules. Compared ALIGN w/o token and ALIGN, we find that the token-level alignment indeed has a positive improvement for the performance.
>
> > Miscellaneous issus
>
> We thank the reviewer's careful reading. We will correct those typos in the revision.
>
> > this paper's method still demands substantial GPU memory
>
> We would like to note that not only the proposed method but also its baseline methods load the pre-trained CLIP as the image and text encoders and thus all compared methods in this paper demand substantial GPU memory. It is a common limitation among the prompt-tuning methods and is beyond the scope of this paper.
>
> >  The method suggested does not appear to be well-suited for a fully zero-shot scenario where there is no training samples
>
> We agree with the reviewer's concern. In fact, all compared baselines (except for CLIP) introduce to-be-learned embeddings for better contiguous prompt tuning, and thus the training samples are needed to train such embeddings. Besides the few-shot and Base-to-New settings, we conduct the cross-dataset experiment, where the models are trained on a source dataset and tested on a target dataset. This allows no overlap categories between the source and target datasets and somewhat can be viewed as a fully zero-shot setting for the target dataset. Our proposed model outperforms baseline methods in most cases, showing the generalization of the method.

---

> > ### Comment · Reviewer_hEEC · 2023-08-21
> > **Response to Rebuttal**
> >
> > Thanks for addressing my feedback; I'm inclined to increase my score from 5 to 6 in favor of the paper's acceptance.

---

> > > ### Author Response · Authors · 2023-08-22
> > > **Thank you**
> > >
> > > We thank Reviewer hEEC for your replies and for increasing your score. Your appreciation encourages us to improve the submission in the revision. Thank you again!

---

### Official Review · Reviewer_S87N · 2023-07-10

**Soundness:** 2 fair
**Presentation:** 3 good
**Contribution:** 2 fair
**Rating:** 4
**Confidence:** 4

**Summary:**

The paper proposes a prompt-learning method for adapting CLIP to few-shot classification. The proposed prompt is multimodal, i.e., both vision and language encoder is adaptable, and multimode, i.e., each modality is assigned with several prompts for diverse representation. Experiments are conducted on few-shot and base-to-new transfer settings, showing the method's superiority over previous state-of-the-art.

**Strengths:**

* Promising results are achieved, both in the few-shot setting and base-to-new transfer setting.
* The presentation is mostly clear and easy to understand.


**Weaknesses:**

* Incremental novelty of the paper. Visual-language prompt learning with multimode prompts and optimal transport (OT) is explored in PLOT [17]. Also, multimodal prompts are explored in works like [27][28]. This proposed multimode multimodal prompts with OT multimode seems to combine the two types of previous methods and reveals limited insights.
* Missing comparison on computation complexity. As multimode prompts require quite large parameters and computations compared to a single prompt, I want to see the comparison of parameter efficiency and time efficiency.
* It is strange that there is no ablation study of each component of the models. It is important to see the individual contribution of the multi-mode prompts and the token-level alignment.

**Questions:**

* In Fig. 3, ALIGN performance on UCF is much superior to previous methods, but the superiority on UCF is not obvious in the Base-to-New setting, especially on the base classes.  Why is that case?
* In Page 3 Line 97, "Empirical findings" needs to specify the source of information or citing papers.

**Limitations:**

Yes

---

> ### Author Rebuttal · Authors · 2023-08-10
>
> We thank reviewer S87N for the comments and suggestions. Below, we address the concerns raised in your review point by point. Please let us know if you have any further concerns or whether this adequately addresses all the issues that you raised with the paper.
>
> > Incremental novelty of the paper
>
> First, we would like to note that building a unified prompt tuning framework that enjoys the good properties of multi-mode and multi-modal methods is novel. It is not the case that it is a trivial combination(e.g., extend MPTs by learning multiple prompts and then apply OT to align two modalities). Technically, the introduced hierarchical OT (prompt-level and token-level OT) in this paper is different from that in PLOT in terms of formulation, motivation, and alignment strategies.
>
> 1) The prompt-level OT models the visual and textual representations as two empirical distributions over the M and N modality-specific prompt embeddings, and the token-level OT further views each prompt embedding as a discrete distribution over its token points (patch or token). Those two formulations enable the proposed model to capture hierarchical features and make fine-grained alignments. However, PLOT calculates the OT distance between the N textual prompts and a set of visual patch embeddings, ignoring the textual token-level alignment, which is different from our cases.
>
> 2) As discussed above, we aim to align two domains hierarchically, the token-level OT focuses on image patches and sentence tokens, and the prompt-level OT focuses on global image and sentence features. This distinguishes ALIGN from PLOT. PLOT focuses on global sentence features and local image patches.
>
> 3) One of the main challenges is to combine the two OTs efficiently. Here we naturally formulate the prompt-level and token-level alignments as a hierarchical OT problem, where the transport distance of the token-level OT acts as the cost matrix of the prompt-level OT, boosting the connection of those two OTs.
>
> Our proposed model belongs to MPTs and provides a new hierarchical OT perspective to improve their performance. In fact, the existent two MPTs [27][28] share a similar idea that is based on traditional continuous prompt tuning. This paper proposes a novel unified framework that can learn multi-mode multi-modal prompts for MPTs and achieves consistent improvement over the existent MPTs.
>
> > Missing comparison on computation complexity
>
> We have reported the detailed comparison in the global comment section. Please check for more discussion.
>
> > Ablation study of each component of the models.
>
> |  Methods  | UCF 101 | Stanford Cars | Flowers102 | Oxford Pets |
> |:--:|:---:|:--:|:----:|:---:|
> | ALIGN w/o prompt |  80.84  | 74.18 |  82.64 | 96.51  |
> |  ALIGN w/o token |  81.04  |  75.32  |  83.09 | 96.61  |
> |   ALIGN   |  81.27  |  76.80  |  83.75   | 96.79 |
>
> Following your advice, we have reported the ablation results of the introduced modules above. We find that both the multi-mode prompts and the token-level alignment contribute to the improvements.
>
> > In Fig. 3, ALIGN performance on UCF is much superior to previous methods.
>
> Here we thank reviewer S87N for pointing out this mismatch case in Fig3. We have checked the numerical results carefully of Fig3 and found that there is typing error on UCF101 16-shot result (85.69 to 95.69). Note that the corrected results still outperform baselines on all few-shot settings.  We apologize again for making you confusion. In terms of the Base-to-New setting, CoCoOp found that CoOp usually overfits the seen set and is not generalizable to the unseen set (higher base-set accuracy and lower new-set accuracy). The proposed model balances the seen and unseen sets well and achieves the highest H score.
>
> > Citing papers at line 97
>
> We will cite related papers [1,2] in the revision.
>
> [1] Yuhang Zang, et al. Unified vision and
> language prompt learning.
>
> [2] Muhammad, et al.  Maple: Multi-modal prompt learning. In CVPR 2023.

---

> ### Author Response · Authors · 2023-08-18
> **Following up with Reviewer S87N**
>
> Dear Reviewer S87N,
>
> We deeply appreciate your thoughtful review and your time. Following your constructive suggestions, we have discussed the different between our method and previous baselines, updated the comparison of learnable parameter and inference time, reported the missed ablation studies, and addressed the typos and citations.
>
> We tried our best to address your concerns, we would like to know if you have any other questions about our paper, and we will be more than happy to have a discussion with you in the following days. If our response has addressed your concerns, would you mind considering re-evaluating our work based on the updated information?
>
> Best regards,
>
> Authors

---

> ### Comment · Senior_Area_Chairs · 2023-08-21
> **final discussions**
>
> Dear Reviewer,
>
> As discussions come to an end soon, this is a polite reminder to engage with the authors in discussion.
> Please note we take note of unresponsive reviewers.
>
> Best regards,
> \
> SAC

---

### Official Review · Reviewer_TnsF · 2023-07-10

**Soundness:** 2 fair
**Presentation:** 2 fair
**Contribution:** 3 good
**Rating:** 5
**Confidence:** 4

**Summary:**

This paper argues that existing prompt turning fails to caption the sample diversity, learning to sub-optimal prompt discovery. To this end, they propose a multi-mode token-level alignment for multi-modal prompt tuning. Specifically, they formulate the prompt turning as the hierarchical optimal transportation problem (distribution matching problem). As a result, the extensive experiments on few-shot image classification, transfer learning, and domain generalization, show the superiority of the ALIGN.

**Strengths:**

1. **The proposed method is effective and comprehensive**. Learning and aligning a set of prompt tokens across modalities by hierarchical optimal transportation is effective and simple.
2. **The result is strong and the evaluation is comprehensive**. The extensive experiments on 15 widely used image datasets under the setting of four task settings show the superiority of the proposed method.

**Weaknesses:**

1. **Time Complexity Analysis.** Since this method introduces hierarchical optimal transmission, I would like to ask whether there is more time overhead in the inference phase than in other methods. Can you provide relevant experimental comparisons in detail?

2. **More Cases for Visualization.** In order to prove that learned prompt tokens have the ability to capture diverse visual concepts, it seems that Fig. 4 cannot explain the above claim well. I am more curious whether there are multiple prompts that will pay attention to the same visual concepts. Can you provide more cases for analysis?

3. **The different with PLOT[1] and MAPLE[2].**  Can you provide a more thorough comparison to illustrate the novelty of your work?

4. **The writing is need to be improved**. It is hard to understand the core idea from the messy introduction. What's more, Sec.2 Background and Sec.4 Related work have a large number of duplicates that exist and can be merged.


[1] PLOT: Prompt Learning with Optimal Transport for Vision-Language Models. ICLR 2023.

[2] MaPLe: Multi-modal Prompt Learning. CVPR 2023.




**Questions:**

As shown in weaknesses.

---

> ### Author Rebuttal · Authors · 2023-08-10
>
> We thank reviewer TnsF for providing positive feedback and helpful suggestions. Below please find a response.
>
> > Time Complexity Analysis
>
> We have provided a detailed experimental comparison at the global comment. Please refer to the above analysis for more details.
>
> >More Cases for Visualization
>
> One of the main motivations of this paper is to mine diverse visual concepts, and thus we formulate the priors of both visual and textual prompt embeddings as the Uniform distributions in Eq(3) of the manuscript, e.g., each visual patch has an equal probability of being attended to. This mathematically guarantees that the learned prompts have the ability to align diverse visual concepts. Empirically, we find that most prompts find their concepts and few prompts attend to the same concepts.
>
> Following your advice, we added more visualization in the uploaded pdf file. Please have a check. We will add those results in our revision.
>
> >The different with PLOT and MAPLE
>
> Compared to PLOT (multiple modes and single modality) and MAPLE (single mode multiple modalities), our proposed multi-mode multi-modal token-level alignment algorithm acts as a unified framework for prompt tuning, e.g., PLOT and MAPLE can be viewed as our particular cases. We briefly summarize the main differences below.
>
> 1) Multi-mode prompt learning for both visual and textual modalities. Our method learns M visual and N textual prompts for two modalities of CLIP, while PLOT learns N textual prompts and MAPLE learns a single visual prompt and single textual prompt. This enables our model to capture diverse visual and textual concepts.
>
> 2) Token-level alignments of both visual and textual modalities. The proposed model aligns the vision and language domains by considering token-level features, e.g., the visual patch features and the textual token embeddings, resulting in fine-grained comparison. The MAPLE and PLOT methods either only consider the prompt-level alignments or focus on visual patch features, failing to model the token-level features of both modalities.
>
> 3) Hierarchical transport framework for prompt tuning. Technically, we formulate the prompt tuning task as a hierarchical OT problem, where the visual and linguistic representations are modeled as the empirical distributions over the M visual and N textual prompt embeddings (prompt-level OT), and the prompt embeddings are further modeled as the empirical distribution over the patch and token features, respectively.
>
> >The writing is need to be improved
>
> Thank you for the writing suggestion, and we will highlight the core idea, clear the Background and Related work section in the revision.

---

### Author Rebuttal · Authors · 2023-08-10

We thank all the reviewers for the time and expertise they have invested in these reviews and for their valuable comments. We are encouraged that all reviewers noted the impressive results achieved by the paper, and reviewers TnsF, hEEC, and Uuq5 praised the novelty and effectiveness of our method. Your comments and suggestions have helped us to improve the paper. We provide a response and clarifications below for each reviewer respectively and hope they can address your concerns.

>Complexity Analysis

A common concern among reviewers is the complexity analysis of the proposed paper. Here we report the comparisons on the number of trainable parameters (#Parameters) and inference time below (FPS):

|  Methods |  CoOp | CoCoOp |   VPT  |  PLOT |    UPT    |   MAPLE   |   ALIGN   |
|:---:|:--:|:--:|:---:|:---:|:--:|:--:|:---:|
| #Parameters | 2,048 | 35,360 | 13,824 | 8,192 | 3,555,072 | 3,555,072 | 3,582,720 |
|  FPS |  645  |   37   |  152  |  583  |  -   | 282  |   62    |

We find that the overall multimodel prompts tuning methods (last three) require more trainable parameters and inference time than single-modal methods.
The proposed ALIGN method aims to learn multi-mode prompts across visual and textual modalities and achieves consistent improvements over baseline methods in most cases. The proposed ALIGN requires slightly more training parameters than UPT and MAPLE because of the multiple prompts. And it also requires more inference time than MAPLE, due to the hierarchical OT operations. The proposed model supports GPU parallel inference and thus has a faster testing time than CoCoOp.

Here, we would like to note that the main idea is to develop a unified prompt tuning framework and provide a new hierarchical OT view for the community. We thank the reviewers for pointing out the potential future work. We will keep this in mind and Continuously improve our method.

---

> ### Author Response · Authors · 2023-08-15
> **The inference time is improved to 129 fps via the parallel computing trick**
>
> We acknowledge all the reviewers for their valuable comments and suggestions, which help us improve the submission. We in this paper propose a unified multi-modal prompt tuning framework that aligns M visual prompts and N textual prompts via the prompt-level and token-level OT distance.  For the commonly concerned issue of complexity analysis, we have reported the number of learnable parameters and inference time of our ALIGN and baseline methods on Food101 datasets at the first rebuttal stage. The proposed model needs to calculate the hierarchical OT distance for each image and label pair and thus has a higher test time (62 fps). To optimize the test time, we empirically find that these OT operations are independent and can be calculated parallelly. This results in a 129 fps inference time, much faster than the previous result.
>
> We thank the reviewers for their constructive suggestions again and believe this inference time improvement widens the practical application of the model. In addition, we would be happy to have further discussions with you, do not hesitate to contact us if you have any questions.

---

### Decision · Program_Chairs · 2023-09-21

**Decision:**

Accept (poster)

**Comment:**

The paper explores a multi-modal prompt learning approach where multiple modes are modeled across modalities. The reviewers have overall appreciated the novelty and experimental gains demonstrated by the authors but also mentioned concerns on compute complexity, experimental comparisons, ablation on components, and clarity on differences with previous works. The author's response helped clarify most of the questions, especially regarding the ablation, compute complexity, and additional results with a larger model.

AC finds the paper interesting and believes it will be a useful result to the community (though components of the proposed approach have been individually used in literature e.g., the hierarchical OT and multi-modal prompt learning, the overall approach shows promise for multi-modal prompt tuning). Therefore, AC recommends acceptance.

However, AC strongly emphasizes on highlighting/correcting the below points in the revision: a) The complexity of the approach is much higher than competitors due to learning multiple-mode prompts and applying hierarchical OT for each image-text pair. This analysis should be included in the paper. b) The visualization provided with rebuttal is poorly labeled and does not really align with the claim "Empirically, we find that most prompts find their concepts and few prompts attend to the same concepts." Authors should check if this can be improved or clarified by explaining the differences. c) Regarding point 1 under differences between PLOT and MAPLE, note that the statement "MAPLE learns a single visual prompt and single textual prompt" is incorrect. b) The clarity in the introduction needs to be improved and the typos need to be fixed in the final revision.